



**To what degree the geometry and kinematics of accretionary wedges in analogue**
**experiments is dependent on material properties**
Ziran JIANG [a], Bin DENG[a*], Caiwei FAN[b], Yu HE[a], Dong LAI[a], Shugen LIU[a], Xinjian WANG[a], Luba
JANSA[c]
a-State Key Laboratory of Oil and Gas Reservoir Geology and Exploitation/Chengdu University of Technology,
Chengdu, China, 610059
b- CNOOC Ltd.-Zhanjiang, Zhanjiang, 524057
c-Earth Science Department, Dalhousie University, Halifax, N.S. Canada.
*Corresponding author: dengbin3000@163.com
**Abstract**: Cohesion and friction coefficients are fundamental parameters of granular materials used in
analogue experiments. Thus, to test the physical characteristics and mechanical behaviour of the
materials used in the experiments will help to better understand into what degree the results of
experiments of geological processes depend on the material properties. Our test suggests significant
differences between quartz sand and glass bead, in particular the shape factors (~1.55 of quartz sand
to ~1.35 glass bead, angular to rounded) and grain sorting (moderately to well sorted). The glass
beads show much better grain sorting and smaller shape factors than the quartz sand. Also they have
smaller friction coefficient (~0.5 to ~0.6) and cohesion (20-30 Pa to 70-100 Pa), no matter of the grain
size in our tested samples. The quartz sand shows much smaller friction coefficient (~0.6 to ~0.65),
and smaller cohesion (~70 Pa to ~100 Pa) than that of smaller grain size sand. We have conducted
four sets of analogue experiments with three repeats at the minimum. Our models show that material
properties have important influence on the geometry and kinematics of the accretionary wedge.
Although the difference in geometries are small, models with larger grain size develop wedges with



higher wedge height, larger taper, shorter wedge length and less number of faults under the same
amount of bulk shortening. In particular, models with basal detachment (even with 1 mm thickness),
show significant difference in geometry and kinematics with that of quartz sand. We thus argue that
the geometry and kinematics of the wedge appear to be significantly influenced by relative brittle and
ductile strengths, and, to a lesser degree by the layering anisotropy. The basal detachment (even of
tiny thickness) determines the first-order control on the location and development of accretionary
wedge, in a contrast to the physical properties of brittle materials.
**Key words:** material property, basal detachment, accretionary wedge, analogue experiment.
**1 Introduction**

Analogue experiments have been used to understand kinematic and dynamic evolution of the

crust, or lithosphere structures, for more than two centuries (e.g., *Hall, 1815, Cadell, 1888*).
Significant progress was made with improvement in monitoring equipment, e.g., X-ray techniques
(*Colletta et al., 1991*), PIV/DIC system (*Adam et al., 2005*).

However, the reproducibility of analogue results and human factor are always suffered in Earth

Science community since then (e.g., *Paola et al., 2009; Graveleau et al., 2012 and references in*).
*Schreurs* et al. (*2006*) suggest that variations in the geometry and evolution of the accretionary wedge
models is result of difference in modelling materials, experimental set-ups etc. In the recent, the
benchmarks experiments were performed at more than twenty laboratories, in the aim to understand the
variability of analogue results and the limits of model interpretation, with each laboratory using their own
analogue material and apparatus (*Schreurs et al., 2006*), or the same material and procedures (*Schreurs et
al., 2016*), or different algorithms (*Buiter et al., 2016*). All models show consistence in the development of
forward thrust propagation and back thrusting, but significant variations are observed in thrusts spacing,



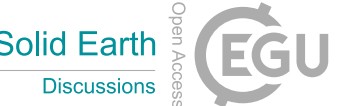

45 their number, surface slope (*Schreurs et al., 2006, 2016; Santimano et al., 2015; Buiter et al., 2016*). These

46 variations show that even small changes in the model setup may affect the mechanical properties of

47 accretionary wedge and thus cause variations in model evolution.

48  Cohesion and friction coefficients are key mechanical parameters in analogue experiments (e.g.,

49 *Lohrmann et al., 2003; Klinkmuller et al., 2016*). To better understand the variability and

50 reproducibility among analogue experiments, we choose simple experiment set-ups of brittle frictional

51 materials, with similar physical characteristics and mechanical behaviour, and focused on accretionary

52 wedge that have been performed in laboratories worldwide (e.g., *Schreurs et al., 2006, 2016; Santimano*

53 *et al., 2015*). It will help to understand to what extent the results of our experiments of geological

54 processes depend on the extrinsic (e.g., model setup, human factor, air humidity etc.,) versus intrinsic

55 variability (e.g., material property, basal friction coefficient and frictional sidewall effect etc.,), which

56 will further help us with meaningful comparisons of models results amongst other laboratories.

57 **2 Material properties**

58 **2.1 Geometry properties of materials**

59  Two kinds of frictional materials have been used in the experiments, e.g., the quartz sand and

60 glass bead, which are used in the laboratories worldwide. At first, all materials are sifted using sieve

61 with sizes of 0.35-0.45 mm and 0.2-0.3 mm. They are divided into four sets to test their geometry

62 properties. The physical characteristics of four sets of materials are summarized in *Fig.1*.

63  The bulk density of each frictional material is estimated by measuring the mass of a known

64 volume, that suggests that the four sets of material have bulk densities of 1.35-1.48 g/cm$^3$ (*Table 1*).

65 Most materials show a unimodal grain size distribution. Two sets of quartz sand, e.g., DB2017-X1 and

66 DB2017-X2, have a roughly homogeneous grain size distribution, with more than 60% of grains



falling within the 0.4-0.7 mm, and 0.25-0.4 mm fractions. Two sets of glass bead, i.e., the DB2017-B1
and B2 have a less homogeneous grain size distribution, with about 50% of the grains falling within
the 0.35-0.6 mm, and 0.3-0.5 mm fractions. The two sets of quartz sand show consistence between the
bulk density and grain size. Samples with the larger grains have higher densities, but the glass beads
are not in this situation.
There is no distinct difference in grain sorting between the quartz and glass beads sand. The
grain sorting of all materials varies from moderately to well sorted. Furthermore, we quantified the
shape of grains using SEM photographic images following the methods of Klinkmuller et al. (*2016*).
Grain shape and outline were measured and averaged from more than 60 grains of each material. The
aspect ratio of four sets of materials varies from 1.34 to 1.56, of which two sets of quartz sand are
characteristics with 1.54 and 1.56, respectively, and two sets glass bead are 1.34 and 1.36, indicating
better grain shape of the latter, as well as of their textures.
**2.3 Mechanical behaviour of materials**
The mechanical properties of the friction materials were determined using Schulze ring-shear
tester at the GFZ in Potsdam, at low confining pressures (0.1-10 kPa) and low shear velocities, similar
to those observed in analogue experiments (*Lohrmann et al., 2003; Klinkmuller et al., 2016*). The
tester consists of a shear cell containing the frictional materials and a lid, the latter is pressed on the
material at given normal load that is constant throughout an experiment. There are sensors at the lid
recording the torque, which can be transformed into shear stress. Ring-shear measurements are
performed at a shear velocity of 3 mm/min for 4 min at a given normal load.
The shear stresses of four sets of materials are shown in *Fig.2*, indicating of varied frictional
properties. At the onset of deformation shear stress increases quickly from zero to a peak level within





a few millimetres of shear (strain hardening phase), and then drops to a stable value (strain softening
phase) that retains for the rest of the deformation until to formation of a shear zone (sliding phase).
When deformation is stopped, the sample unloaded and subsequently deformation is resumed.
Renewed shearing results in a second and similar shear curve, resulting in another stress peak
(reactivation phase). That is distinctly smaller than the first peak level, and roughly larger than the
value of the first stable phase (*Fig.2*). It should be noted that the slightly increased values are artifact
of the setup, result of the fact that the lid of shear cell slowly burrows into the tested materials during
shearing, thereby increasing the friction at its side walls (*Lohrmann et al., 2003*). Furthermore, three
values of friction strengths, e.g., peak strength, dynamic strength and reactivation strength, are picked
manually from these curves, for the applied normal load. For each material, the three values of friction
strengths, e.g., peak strength, dynamic strength and reactivation strength, are determined for six
different normal loads varying between 500 Pa and 16000 Pa. Each normal load step is repeated three
times, resulting in a total of 18 measurements for each material.
Measured values of peak strength, dynamic strength and reactivation strength are plotted against
the applied normal stresses, respectively (*Fig.3*). All four sets of materials show an approximately
linear increase of all three values with normal stresses, consistent with a Mohr-Coulomb failure
criterion. Thus, a linear regression analysis is applied to the three values of all materials, to obtain
their friction coefficient ($u$), which corresponds to the slope of the line and the friction angle ($\tan^{-1} u$).
Furthermore, the cohesion ($C$) is the linearly extrapolated value at zero normal stress (*Table 1*). It
should be noted that the failure envelopes for frictional materials is usually non-linear at low normal
stresses. We use further an alternative method to derive friction coefficients and related cohesion of
four sets of materials. This method calculates two point slopes and their intercepts for mutually





combined pairs of a data set (e.g., *Klinkmuller et al., 2016*). A total of 18 measurements for each
material thus resulted into 135 data sets for friction coefficient and cohesion. Those are then evaluated
by means of calculating mean and standard deviation and comparing the probability density function
to a normal distribution (*Fig.3*).

For the data sets obtained by two methods of the linear regression and mutual pairs regression

analysis, we have found a slight difference between them. (1) peaks of the experimental probability
density function are close to or narrower than a normal distribution. (2) cohesion values from the
mutual pairs regression analysis are usually smaller than the values from the linear regression analysis.
We thus prefer the calculated standard deviation as a conservative value for the four sets of frictional
materials (*Table 1*).

For all the four sets of material, there is a systematic decrease in the values of friction coefficient

from internal peak friction to internal reactivation friction, to internal dynamic friction (*Fig.3*). At the
same way, the angles of them systematically decrease with 2-5 ° by turn (*Table 1*). Internal peak
friction angles are 38 ° for two sets of quartz sand, with friction coefficients of 0.783 and 0.798 (e.g.,
DB2017-X1 and X2), respectively. Glass beads have much lower angles of internal peak friction of
31 °, and friction coefficients of 0.594 and 0.612 (e.g., DB2017-B1 and B2).

Internal reactivation friction and dynamic friction angles for sample DB2017-X1 are 34 ° and 31 °,

with friction coefficients of 0.687 and 0.599, respectively. For sample DB2017-X2 with much smaller
grain size than the former one, those angles are 33 ° and 30 ° with related friction coefficients of 0.656
and 0.582, indicating much smaller values than those of DB2017-X1. Two sets of glass beads have
lower angles of internal reactivation friction and dynamic friction with 28 ° and 25 °, 30 ° and 26 °,
respectively. Whilst the friction coefficients are 0.530 and 0.495, 0.569 and 0.493 for samples of



DB2017-B1 and B2. For the two sets of glass beads, the internal friction angles distinctly increase
with the decreased mean grain size, but not in the quartz sands. It should be noted that the internal
friction angles of glass beads are substantially smaller than that of quartz sands, no matter of their
mean grain size.

The extrapolated cohesion values of internal peak friction, reactivation friction and dynamic

friction vary considerably, in particular the internal peak friction. Sample DB2017-X1 is characterized
by roughly similar cohesion values of reactivation friction and dynamic friction, e.g., 68 Pa,
significantly larger than that of internal peak friction with -9 Pa. For sample DB2017-X2, the
cohesion values of internal reactivation friction and dynamic friction are 125 Pa and 92 Pa, in contrast
to peak 2 Pa of cohesion values at internal peak friction. Extrapolated cohesion values of glass beads
are distinctly smaller than that of poor quartz sand (*Fig.3*). The cohesion values of internal
reactivation friction and dynamic friction are 28 Pa and 16 Pa, 71 Pa and 37 Pa (e.g., DB2017-B1 and
DB2017-B2), respectively. In the four sets of materials, the cohesion value of reactivation friction is
highest, whilst the peak friction is the lowest.

*Klinkmuller et al. (2016)* used the same ring-shear tester to determine the material properties of

frictional materials widely used in more than twenty laboratories worldwide. The obtained values
correspond closely to ours, with internal friction angles of 32-40 ° at peak friction, and mean values of
30-37 °, 28-34 ° at reactivation friction and at dynamic friction, respectively. Most of their values of
friction coefficient at dynamic friction and reactivation friction are roughly equal, and substantially
smaller than that at peak friction.
**3 Experiment setup and results**
**3.1 Experiment setup**



In all experimental set-ups, a quartz sand wedge with horizontal base and surface slope was
sieved in with 48 cm height into the deformation apparatus with an initial sand pack of 800×340×350
mm. Of which color quartz sand with thickness of ~1 mm was used as a layer marker in the
experiments. To reduce the amount of friction, a lubrication of glass wall was done before
sieving-load quartz sand. Thus, there is no significant bias of frictional sidewall effect in our
experiments, as the ratio of the area contacts of the sand body with glass sidewalls to its area of
contact with basement remains 0.05-0.1 (*Souloumiac et al., 2012*). Sand models were deformed in
pure shear by moving a vertical rigid wall from right side with a constant velocity of 0.001 mm/s (e.g.,
*Deng et al., 2017*). After 400 mm shortening , a comparison of all results was carried out.
Although slight difference may be in the material properties, variations in material properties are
important for differences in the geometry and structural evolution of experimental models (*Schreurs et
al., 2006, 2016*), e.g., and kinematics of thrust wedges as a function of their material properties
(*Lohrmann et al., 2003*). To understand how important material properties in our analogue
experiments are, we conducted six experiments with two sets of quartz sand (e.g., No.1 and No.4),
and two sets with glass bead with 1 mm thickness (e.g., No.2 and No.4) and 3 mm thickness (e.g.,
No.3 and No.6) (*Table 2, Fig.4*).
The deformation of wedge was photographically recorded using time-lapse photography at every
1.0 mm of contraction. Using a graphic software package, a set of parameters was systematically
measured at 10 mm intervals to describe quantitative results of the wedge. Cross-sections allow us to
measure the wedge geometies and fault spacing, following the method used by Buiter et al. (*2016*)
and Schreurs et al. (*2016*) in their experiments. In particular, the wedge slope angle was measured as
the best fitting line through the intersection of the fault tips and the surface of accretionary wedge



(e.g.g, *Stockmal et al., 2007*).

## 3.2 Experiment results of quartz sands

At first, we tested each set of quartz sands in a classic analogue experiment, with similar set-up

to analyze the deformation style and mechanical behavior. All results confirm that deformation of
quartz sand generate accretionary wedges with thrust planes dipping toward the moving wall and
propagating sequentially forward (*Fig.4*). However, deformation styles are slightly different between
materials after 400 mm shortening. Setup No.1 and No.4 present few well-individualized thrusts and
back thrusts and low slope angle (18.7 °for No.1 to 17.5 °for No.4). Besides, the setup No.1 has
higher wedge height (135.3 mm to 124.0 mm) and shorter wedge length (292.6 mm to 302.2 mm)
than that of No.4. This is certainly due to its lower cohesion and smaller friction coefficient than the
quartz sand of DB2017-X2, used in the setup No.4.

During progressive shortening, accretionary wedges show common characteristics such as: (1) a

rapid growth and subsequent slow self-similar growth (*Fig.5, Fig.6*), consistent with the critical taper
theory (*e.g., Storti et al., 2000; McClay & Whitehouse, 2004; Deng et al., 2017*), and (2) quartz sand
slides stably and is translated/moved along the horizontal base and is affected by internal deformation
during the self-similar growth processes. All model wedges grows rapidly in height and length with
progressive shortening during the early stage, until a critical wedge state were attained at ~100 mm
shortening (*Fig.6*), at which three (e.g., setup No.1) and four (e.g., No.4) developed in-sequence
imbricate thrusts nucleated and formed an internal backstop. Subsequently, the wedges growth are
self-similar, or quasi-stable. There are sharp jumps in the wedge slope angle and length, that reflect
the nucleation of each new foreland-verging thrust. The subsequent decrease in wedge length prior to
the development of the next new thrust indicates internal shortening and deformation within the





wedge. It should be further noted that there is a slight decrease in the wedge slope angle during the
progressive shortening, followed by a distinct increase in the angle (*Fig.5*). It implies that for the
wedge to overcome the basal and internal friction, it undergoes internal deformation with
layer-parallel shortening until it again reaches a critical wedge slope that brings accretionary wedge
slide and translation foreland.

In the two models with quartz sand, there are distinct changes in the wedge slope angle (e.g.,

3-5 ° for No.1 and 4-6 ° for No.4), and wedge length (e.g., 10-30 mm for No.1 and 10-20 mm for No.4)
during the self-similar growth progress. However, the difference in wedge slope angle between No.1
and No.4 is roughly 2-4 °, and ~10 mm for wedge length with a certain given shortening (*Fig.5, Fig.6*),
indicating similarity in the wedge geometries.
**3.3 Experimental results of quartz sand with basal detachment**

In our model comparison, we choose to use quartz sand and glass bead used at the laboratory,

e.g., setup No.2 and No.3, setup No.5 and No.6. In these four models, all accretionary wedges show a
rapid growth and subsequent slow self-similar growth. After 400 mm shortening, setup No.2 and No.3
present fewer well-individualized foreland thrusts (5 and 6) and lower slope angle (10.3 ° and 9.8 °)
than the setup No.1, as well as shorter wedge height (e.g., 101.9 mm for No.2, 102.0 mm for No.3)
and longer wedge length (e.g., 375.1 mm, 349.3 mm) (*Fig.4*). Furthermore, setup No.5 and No.6 show
fewer well-individualized foreland thrusts (9 and 6) and lower slope angle (16.5 ° and 12.2 °), as well
as shorter wedge height (e.g., 122.2 mm, 106.8 mm) and longer wedge length (e.g., 328.6 mm 327.3
mm) than the setup No.3. In particular, more backthrusts developed in these experiments setup No.1
and No.4, consequently accretionary wedges are characterized by small pop-up structures. We argue
that such variability is due to basal detachment with glass beads in these four experiments. It should



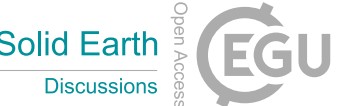

be noted that the wedge slope angle and wedge height decreased with increasing thickness of basal
detachment with glass beads, as well as wedge length increased, e.g., from setup No.1 to No.3, and
No.4 to No.6, respectively.

During progressive shortening, there are sharp jumps in the wedge slope angle and length,

followed by slow decrease of their values in the last, self-similar growth progress (*Fig.5, Fig.6*). It is
consistent with internal deformation of layer-parallel shortening (e.g., *Koyi and Vendeville, 2003;*
*Deng et al., 2017*). For setup No.2 and No.3, there are distinct changes in the wedge slope angle (e.g.,
2-4 °and 4-6 °), and wedge length (e.g., 20-40 mm and 10-30 mm) during the self-similar growth
progress, than at the setup No.1. However, the variations of wedge slope angle and length are 3-5 °and
2-4 °, 10-20 mm and 10-30 mm for No.5 and No.6, respectively. Although no distinct difference of
wedge slope angle is between No.1 and No.4 setup, significant variations occurred between No.2 and
No.5 (e.g., 4-10 °), and between No.3 and No.6 (e.g., 2-8 °) setups (*Fig.5*)**.** Similarly, significant
variations in wedge length can be found between No.2 and No.5 (e.g., 20-50 mm) setups, which are
much larger than those between No.3 and No.6 (e.g., 10-30 mm) setups (*Fig.6*). Thus, we suggest that
the mechanical properties consisted of lower internal friction and cohesion, e.g., glass beads at basal
detachment, will substantially affect the wedge geometry.
**4 Discussion**
**4.1 Wedge geometry with various materials**

That a decrease in wedge strength controlled by internal friction and cohesion of materials, as the

decreases of the slope angle and height, and increases of the wedge length have been proven by
several experiments (e.g., *Koyi and Vendeville, 2003; Nilforoushan et al., 2008*). The topography lines
for each 2 cm shortening have been depicted in all models (*Fig.7*), which shows an increase of the



wedge height during progressive shortening. However, the height of wedges including no
low-frictional basal detachment (e.g., setup No.1 and No.4) constantly increases and hinterland thrusts
are active during all stages of shortening. In wedges including basal detachments (e.g., glass beads),
the forward thrusts are inactive and backthrusts are active in the hinterland zone (*Fig.5*). The height of
wedges remains constant after the deformation is transferred into the foreland zone. Analysis of the
wedge geometry of models (e.g., Nos. 2, 3, 5 and 6), shows that the height of the wedges remains in a
steady state after a certain shortening, e.g., 340 mm shortening for No.2 and No.5, 300 mm shortening
for No. 3 and No. 6, respectively. It suggests that the accretionary wedge slides and is translated along
the basal detachment in a steady state.

We have found, when investigating these models, that the internal friction and cohesion variation

changes the wedge slope angles. However, the difference in geometry of models, using only frictional
materials (e.g., quartz sand X1 and X2), is not distinct. In another way, all wedges used only frictional
materials show a very similarity in the wedge geometry. As we have illustrated previously, the
difference in geometry, e.g. slope angle, number of forward and back thrusts is more pronounced
when models contain basal detachment of glass bead. This implies that the basal detachment
determine the first-order control on localization and development of accretionary wedge, as opposed
to the properties of brittle materials (e.g., *Teixell and Koyi, 2003; Ellis et al., 2004*). We thus infer that
with more complex brittle-viscous rheology, there are more complicated variations in the accretionary
wedge.
**4.2 Wedge kinematics with various materials**

The evolution of all models is roughly similar, with development of accretionary wedge by

in-sequence forward thrusting and by minor back thrusting. In general, thrusts are nucleated soon after



the beginning of shortening at the base of the models. They are propagating upward across the
accretionary wedge and then reach the top surface as a brittle structure. However, significant
variations existed between models in kinematics (*Fig.8*), in particular in the number of thrusts, fault
space and fault displacement (*Ellis et al., 2004; Schreurs et al., 2006, 2016; Santimano et al., 2015*).
During the early stage of deformation, closely forward thrusts developed with regular spacing across
the models. Thus, the fault spacing and displacement are substantially smaller in the early stage than
in the later stage. Subsequently, the kinematic evolution of these models distinctively changes, the
number of thrusts decrease and spacing between successive imbricate thrusts increase significantly.
The imbricate forward thrusts are characterized by comparative fault spacing and displacement. The
important point in these models is that during the progressive shortening, the sequence of thrusts
formation is quite rapid in models with basal detachment, and consequently accommodated with
fewer forward thrusts. The thicker is the basal detachment, the fewer fault number is in the wedge and
vice versa.
Forward thrusts are more frequent and closely spaced with smaller displacement in the earlier
stages of deformation, and widely spaced with larger displacement during later stage of deformation.
However, thrusts, which developed above the basal detachment, are lesser in number and relatively
widely spaced and displaced in all models. In particular, a roughly linear increase of fault spacing can
be found in models with basal detachment, e.g., $D_{(T3/T2)}$ to $D_{(T5/T4)}$ in setup No.3, $D_{(T4/T3)}$ to $D_{(T6/T5)}$
in setup No.5 and No.6 (*Fig.9*), no matter of the thickness of the detachment during the later stage.
It should be noted that glass bead in these models, even with a limited thickness (e.g., ~ 1 mm in
setup No.2 and No.5), acts as basal detachment and triggers minor thrusts with locally modified thrust
trajectories. This is evidenced by (1) development of second order thrusts, e.g., $T_{5-1}$ a nd $T_{5-2}$ in setup



No.2 (*Fig.5*); (2) widespread development of backthrusts, e.g., in setup No.2 and No.5; (3)
development of small ramp and flat geometry, e.g., thrust $T_4$ and $T_5$ in setup No.2 ; (4) variable
displacement and slip along the thrusts, e.g., amount of displacement for each thrusts and slip
measured at the surface being large and decreasing with increasing depth (*Ahamd et al., 2014;*
*Schreurs et al., 2016*).

**4.3 Extrinsic versus intrinsic variability of models**

Both extrinsic and intrinsic variability of analogue experiments have influence on the geometry

and kinematics of accretionary wedges. Therefore, we used statistical analysis to study extrinsic and
intrinsic variability (e.g., material properties) at the stage of self-similar growth of wedges, following
the methods of Santimano et al. *(2015)*. Except the wedge length with values larger 0.2, the statistical
results of coefficient of variation (CV) show that most of parameters range from 0.05 to 0.2, with an
average of ~0.1 (*Fig.9*). Accordingly the CV is lower for thrust-ramp displacement (CV=0.01-0.1),
thrust-ramp angle (CV=0.07-0.18), wedge height (CV=0.06-0.14) and wedge slope (CV=0.05-0.2),
and higher for wedge length (CV=0.2-0.31). The main difference between those parameters is that
wedge length is time dependent and may reflect evolving wedge dynamics, however, the other
parameters are not time dependent, related to properties of the entire wedge.

Furthermore, the statistical test ANOVA shows that parameters can be divided into two categories,

based on their P values and $R^2$ values. For the first category, most of the p values for wedge slope are
much smaller than 0.05, and with larger $R^2$ values > 0.1. The second category is with higher p values
of 0.4-1.0 (most are with values of 0.6-1.0), and lower $R^2$ values < 0.1 (most are with values <0.02).
Accordingly the p value for thrust-ramp angle and displacement are 0.36-0.94, 0.39-0.98, for wedge
length and height are 0.62-0.89, 0.28-0.98, respectively.

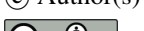



As we known, a large $R^2$ and smaller p values suggest that the variation in our models is due to
the experimental setup, or extrinsic sources, rather than due to the variation with the system (*Zar,*
*2010; Santimano et al., 2015*). In particular, a p value >5% suggests repeatability of the data from
different experiments of the same setup, or reproducibility of the model. Therefore, the statistical test
ANOVA recognizes that the variation in the observables (e.g., geometry and kinematic parameters) is
repeatable between our analogue experiments, except for the wedge slope. It further indicates
increased effect on the wedge slope angle from the extrinsic variability, e.g., human-factor, or more
susceptible to extrinsic changes in the accretionary wedge (*Buiter et al., 2006; Santimano et al.,*
*2015*).
**4.5 Comparison of the Natural Examples**
In addition to the investigation of the effect on mechanical properties of accretionary wedge, we
can consider the role of weak basal detachments on the geometry and deformation in natural examples,
like the Zagros fold-thrust-belt and Longmenshan fold-thrust-belt. In our models, we observe that the
different thickness of quartz sand above weak basal detachment deforms differently. The upper
frictional material decouples the deformation, and the geometry and kinematics of structures above
the basal detachment are different. A similar deformation mechanism was reported by Sherkati et al.
(*2006*), who used surface and seismic data and borehole information to construct interpreted
cross-section of the Zagros. They suggested that the deformation across and along the Zagros belt
varies due to the spatial distribution of shale and evaporitic layers. Such geometrical and kinematic
changes are further supported by analogue experiments that related to different parts of the Zagros
belt (*Sherkati et al., 2005; Deng et al., 2017*).
Another natural example is from the Longmenshan fold-thrust-belt at eastern margin of Tibetan





plateau, where there is significant change in the thickness of Lower Cambrian Qiongzhushi Formation,
dominated by black shale. The thickness of Qiongzhushi Formation is at maximum of ~1500 m, in a
contrast to ~ 0 m in the southern segment of the western foreland basin, as result of the erosion (*Liu et*
*al., 2017*). During the Late Triassic, the Songpan-Ganzhi flysch strata were thrust southeastward onto
the Sichuan Basin, along the Longmenshan fold-thrust-belt, to form the western Sichuan foreland
basin (*Li et al., 2003; Liu et al., 2012*), as an accretionary wedge. The structural configuration across
the Longmenshan fold-thrust-belt is shown in cross-section that has been constructed using seismic
reflection profiles and borehole data (e.g., *Jia et al., 2006; Lu et al., 2012*). In the northern segment of
the Longmenshan the Palozoic strata, such as at Tianjingshan and Anxian areas, was southeastward
thrusted onto the gentle deformed Mesozoic strata in the foreland basin (*Jing et al., 2009; Lu et al.,*
*2012*). In particular, there was substantial increase in the thickness of the anticline core comprised by
Mesozoic strata, due to shortening deformation of the Lower Cambrian strata. The deformation of
Mesozoic strata on anticlinal limbs reveals contemporaneity of tectonic activity. In the profile, the
deep-seated strata are associated with pop-up structures, as shown e.g., in the Well Tianjian-1, and
almost all the thrusts are associated with minor backthrusts. Such a structural style shows close
similarity with one observed in our models (*Fig. 5*). To the southern segment of the Longmenshan, the
main structural features are dominated with prominent thrusts that rooted in the base, probably in
Sinian units (*Jia et al., 2006; Hubbard et al., 2010*). Similar feature was observed in the analogue
experiments with high-friction basal detachment. Such correlation between deformation with basal
detachment is further associated with different topography and slope across the Longmenshan
fold-thrust-belt, e.g., much higher topography and slope in the southern segment of Longmenshan
than that of northern segment (*Kirby and Ouimet, 2011; Li et al., 2012*).





In addition to influencing the geometry and kinematics of model wedges, the basal detachment
also governs both the volumetric-strain and layer-parallel shortening of the wedge (*Teixell and Koyi,*
*2003; Nilfouroushan et al., 2012*). Applied to the nature, our model results suggest that more forward
and back thrusts and deformation with higher volumetric-strain are expected in convergent settings,
with a high-friction basal detachment, than in those shortened above low-friction basal detachment, or
a weak base. Such deformation has major implications for prospecting hydrocarbon systems within
fold-and-thrust belts.
**5 Conclusion**
In analogue experiments as well as in the nature, material properties and mechanical stratigraphy
are important elements in geometry and kinematics of accretionary wedge. Its evolution shows a rapid
growth and subsequent slow self-similar growth, that wedge slides and is translated along the
horizontal base in a steady state. However, the material properties affect the wedge geometry and
kinematics in various ways. Two setups of models with quartz sand show no distinct difference in
wedge geometry, however, model with larger grain size developed wedge with distinct variations in
wedge kinematics. In particular, models with 1 mm thick glass beads bed show significant differences
from experiments with quartz sand, e.g., lower wedge height and smaller taper, shorter wedge length
and less number of faults. The changes in the geometry and kinematics of accretionary wedge are
most pronounced when the thickness of basal detachment is larger.
Applied to the nature, our model results suggest that more forward and back thrusts companied
with lower wedge slope angle and height and larger wedge length, are expected in convergent settings
with a high-friction basal detachment, than in those shortened above a low-friction basal detachment,
e.g., the salt formation under parts of the Zagros fold-thrust belt, and shale formation under parts of





the northern segment of the Longmenshan fold-thrust belt.

**Acknowledgements**

This work was supported by the Natural Science Foundation of China (Nos. 41572111) and Natural
Science Foundation of Sichuan Province (No. 2017JQ0025), and by EPOS TAN and its member GFZ
Helmholtz Centre POTSDAM, and we appreciate Dr. Matthias Rosenau for his help in the ring-shear
testing and manuscript.

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



**Figure Captions**


**Fig.1 Physical characteristics of quartz sand and glass bead used in the experiments. Upper and**
**lowermost pictures are SEM images, respectively.**
**Fig.2. Shear stress plotted as a function of cell displacement (~the amount of shear strain) for quartz**
**sands (X1 and X2) and glass beads (B1 and B2) for six different normal loads (500, 1000, 2000, 4000, 8000**
**and 16000 Pa).**
**Fig.3. Ring-shear test data analysis (four sets of materials): on the left is linear regression analysis of**
**shear strength (peak, dynamic, reactivation) vs. normal load data pairs (18 data); on the right is**
**histograms of friction coefficients and cohesion derived from mutual two-point regression analysis (135**
**data).**
**Fig.4 Photographs of six experiments and their interpretations. Setups 1 and 4 are of quartz sands with**
**mean grain sizes of 0.54 mm and 0.34 mm, respectively, setups 2-3, and setups 5-6 are of quartz sand and**
**glass bead. The interpretation suggests significant change in structures due to the presence of glass beads**
**in the model setup.**
**Fig.5 Plot of the wedge slope angle of accretionary wedge versus shortening displacement. The slope angle**
**decreases episodically with the formation of a new thrust in each model, however, it remains roughly**
**constant after attaining critical wedge at 100-150 mm shortening.**
**Fig.6 Plot of geometries (e.g., the wedge length and height) of accretionary wedge versus shortening**
**displacement. The wedge geometries show significant changes in wedge length and height with increasing**
**shortening velocities. The length increases episodically with the formation of new thrust in each model,**
**however, angle and height remain roughly constant after attaining a critical wedge.**
**Fig.7 Topography lines are depicted in the models for each 2 cm of shortening.**





**Fig.8 Fault spacing and displacement to show different kinematics in the models, (D($_{T1/T2}$) indicates the**
**fault spacing between the forward thrust T1 and T2).**
**Fig.9 Plots showing the (a) p value (ANOVA test) and (b) $R^2$ (ANOVA test) and (c) coefficient of variation**
**for each setups.**
**Tables**
**Table 1 Physical characteristics of tested granular materials**
**Tables 2 Geometries of accretionary wedges with tested materials**





**Table 1 Physical characteristics of tested granular materials**

| Sample No. | Grain size mm | Mean grain size mm | Density g/cm³ | Grain sorting | Shape factors | | Dynamic | | | Reactivation | | | Peak | | |
|---|---|---|---|---|---|---|---|---|---|---|---|---|---|---|---|
| | | | | | Angular /rounded | Aspect ratio | Friction coefficient | Angle | Coefficient (Pa) | Friction coefficient | Angle | Coefficient (Pa) | Friction coefficient | Angle | Coefficient (Pa) |
| DB2017-X1 | 0.3~0.45 | 0.543 | 1.43 | MW | A | 1.54 | 0.599 | 30.922 | 68.527 | 0.687 | 34.489 | 101.630 | 0.783 | 38.061 | -9.740 |
| DB2017-X2 | 0.2~0.3 | 0.342 | 1.35 | MW | A | 1.56 | 0.582 | 30.199 | 92.299 | 0.656 | 33.265 | 124.949 | 0.798 | 38.590 | 1.869 |
| DB2017-B1 | 0.3~0.45 | 0.448 | 1.37 | M | R | 1.34 | 0.459 | 24.655 | 16.001 | 0.530 | 27.924 | 28.516 | 0.594 | 30.710 | -56.643 |
| DB2017-B2 | 0.2~0.3 | 0.371 | 1.48 | MW | R | 1.36 | 0.493 | 26.243 | 37.014 | 0.569 | 29.640 | 71.082 | 0.612 | 31.467 | 16.364 |

**Tables 2 Geometries of accretionary wedges with tested materials**

| Setup | Materials | Wedge height (mm) | Wedge length (mm) | Wedge taper (°) | Fault numbers (n) | Fault spacing (mm) |
|---|---|---|---|---|---|---|
| No.1 | X1 | 135.3 | 292.6 | 18.7 | 8 | 8.2~110.2 |
| No.2 | X1+B2 (1 mm) | 101.9 | 375.1 | 10.3 | 10 | 12.6~73.5 |
| No.3 | X1+B2 (3 mm) | 102.0 | 349.3 | 9.8 | 5 | 224~108.1 |
| No.4 | X2 | 124.0 | 302.2 | 17.5 | 9 | 5.6~95.0 |
| No.5 | X2+B1 (1 mm) | 122.2 | 328.6 | 16.5 | 10 | 11.6~49.6 |
| No.6 | X2+B1 (3 mm) | 106.8 | 327.3 | 12.2 | 7 | 14.8~93.1 |




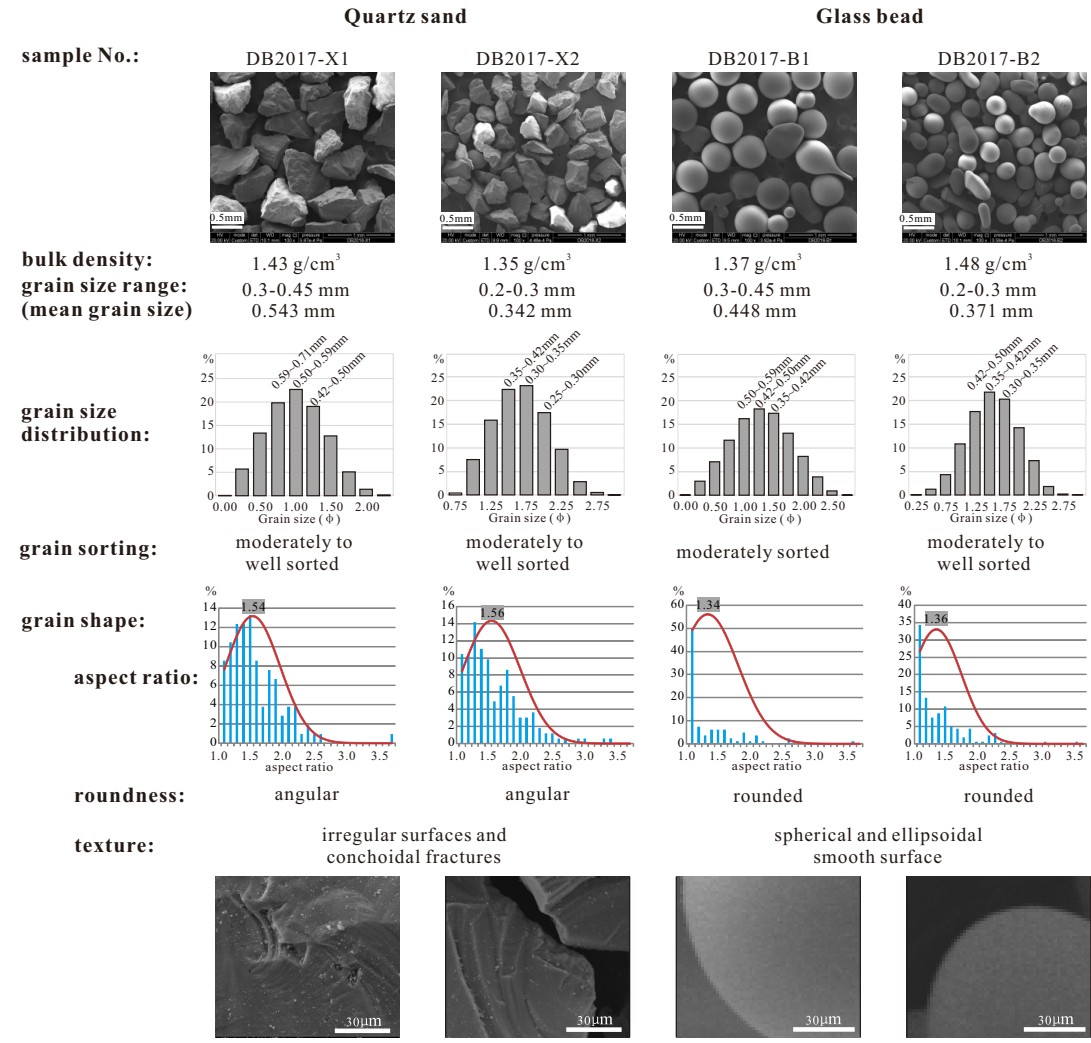





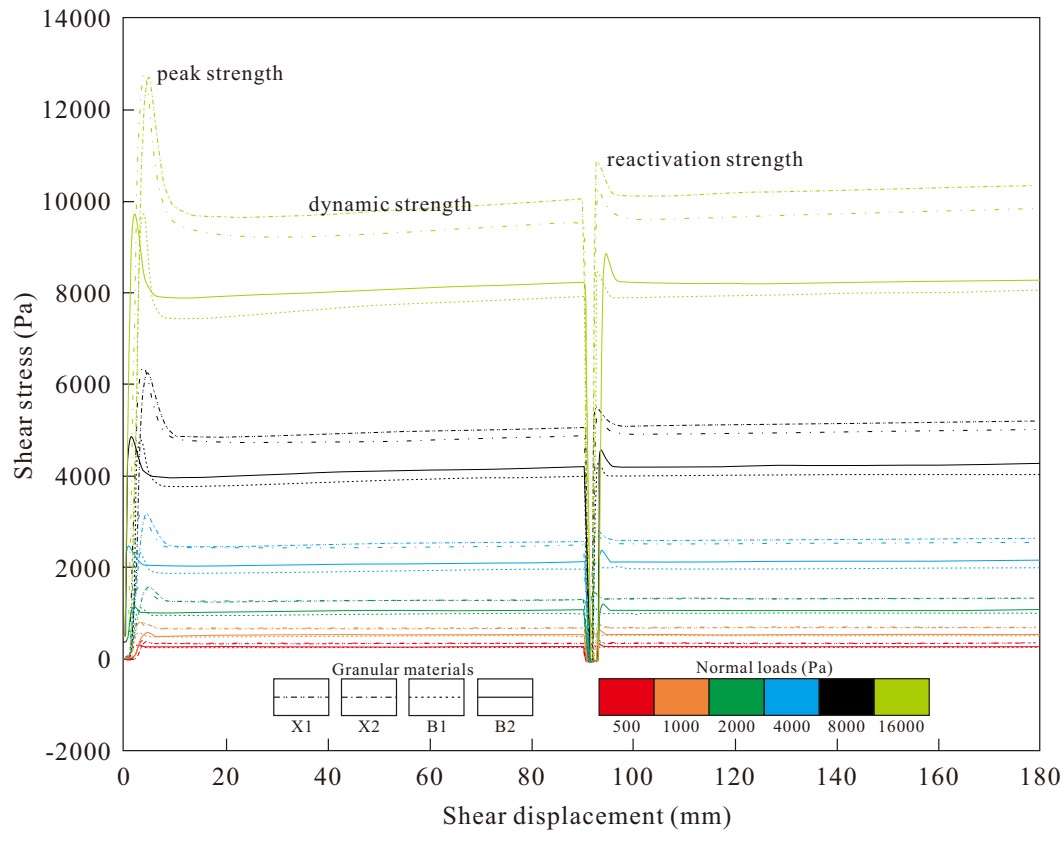





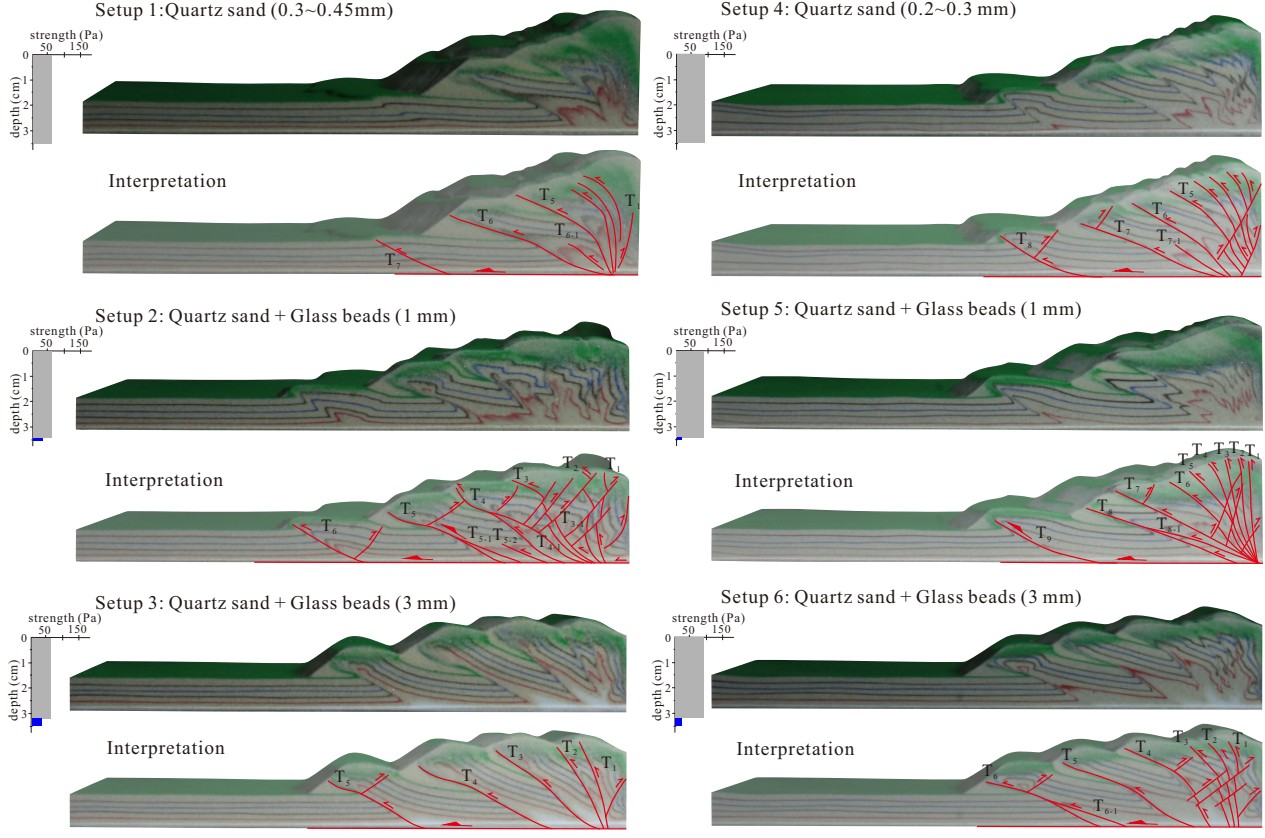





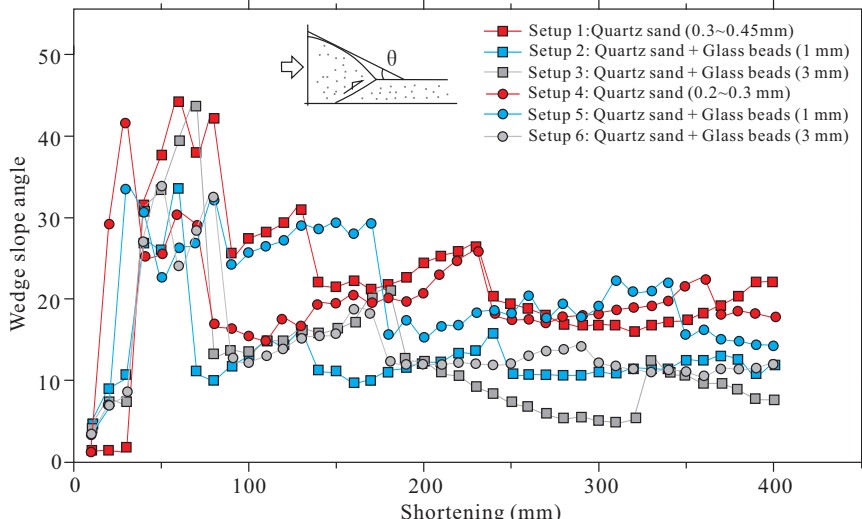





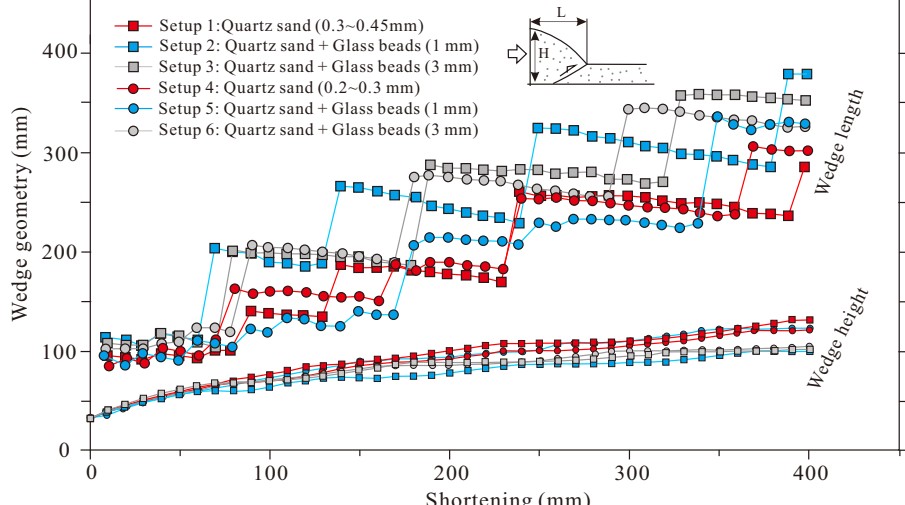





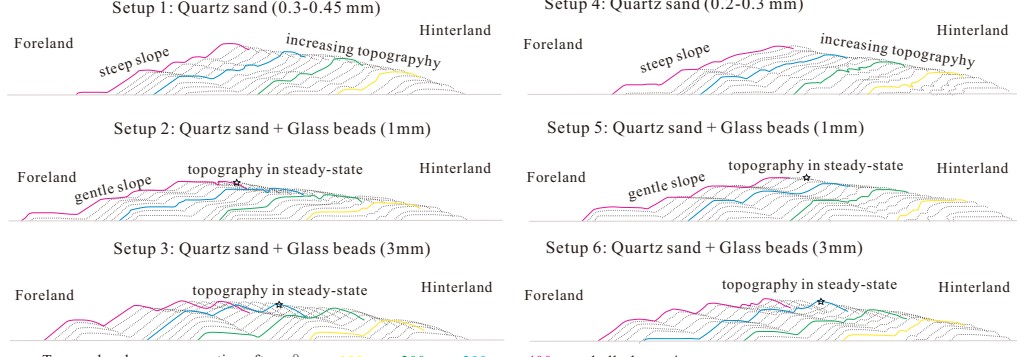

Topgraphy along cross section after: 0 mm, 100 mm, 200 mm, 300 mm, 400 mm, bulk shortening.




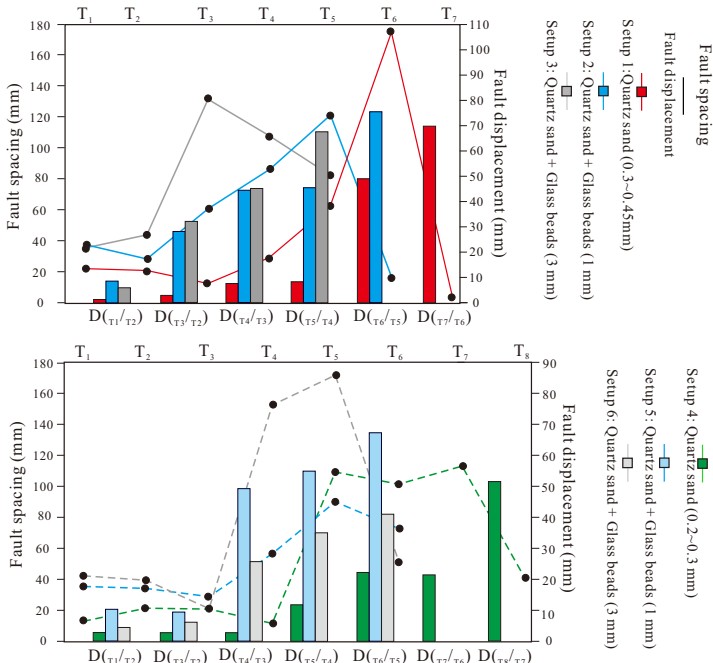





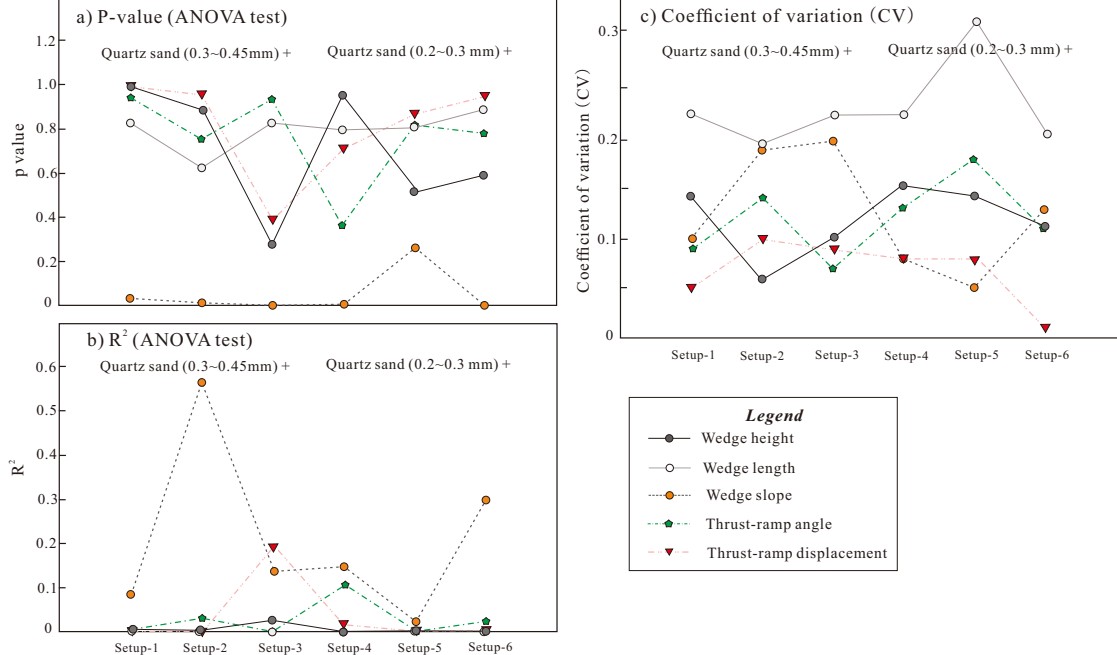