# Peer review of "To what degree the geometry and kinematics of accretionary wedges in analogue"

_Solid Earth, 2018_

## Short Comment (SC1) · 14 Aug 2018

[revised manuscript text omitted]

---

## Author Comment (AC1) · 14 Aug 2018

Supplement to "To what degree the geometry and kinematics of accretionary wedges in analogue experiments is dependent on material properties". Deng , B. Rosenau, M., Schönebeck, J. (2018): Ring-shear test data of rock analogue materials from Chengdu University of Technology (EPOS Transnational Access Call 2018). GFZ Data Services, http://doi.org/10.5880/GFZ.4.1.2016.008.

Please also note the supplement to this comment:
https://www.solid-earth-discuss.net/se-2018-45/se-2018-45-AC1-supplement.pdf

[Figure]

**Supplement:**

**Ring-shear test data of rock analogue materials from Chengdu University of Technology (EPOS Transnational Access Call 2017)**
(http://doi.org/10.5880/GFZ.4.1.2018.---)

**Citation**

This data is freely available under a Creative Commons Attribution 4.0 International (CC-BY 4.0) Licence. When using the data please cite is like:

Deng , B. Rosenau, M., Schönebeck, J. (2018): Ring-shear test data of rock analogue materials from Chengdu University of Technology (EPOS Transnational Access Call 2018). GFZ Data Services, http://doi.org/10.5880/GFZ.4.1.2016.008

**The data are supplementary material to:**

Jiang, Z., Deng, B., Fan, C., He, Y., Lai, D., Liu, S., Wang, X., Jansa, L.: To what degree the geometry and kinematics of accretionary wedges in analogue experiments is dependent on material properties. Solid Earth Discussions,  https://doi.org/10.5194/se-2018-45

**Data Description**

This dataset provides friction data from ring-shear test (RST) on natural and artificial granular materials used for analogue modelling in Jiang et al. (2018) and in the experimental lab of Chengdu University of Technology (CDUT, China).  Six samples, 4 quartz sands and 2 glass beads, have been characterized by means of friction coefficients and cohesion. The material samples have been analysed at GFZ Potsdam in the framework of the EPOS (European Plate Observing System) Transnational Access (TNA) call of the Thematic Core Service (TCS) Multi-scale Laboratories (MSL) in 2017 as a remote service for CDUT.

A first application of the materials tested as well as further details of the materials, measuring techniques as well as interpretation and discussion of results can be found in Jiang et al. (2018) to which this dataset is supplement material.

**1. Measurement procedure:**

The data presented here are derived by ring shear testing using a SCHULZE RST-01.pc (Schulze, 1994, 2003, 2008) at the laboratory for tectonic modelling of the Helmholtz Centre Potsdam (GFZ Potsdam, HelTec). The RST is specially designed to measure friction coefficients and cohesion in loose granular material accurately at low confining pressures and shear velocities similar to sandbox experiments. In this tester, a sand layer is sheared internally at constant normal load and velocity while the shear stress and volume change is measured continuously. For more details see Klinkmüller et al. (2016) and Ritter et al. (2018).

Each sample has been carefully prepared by the same person and measured consistently following the same protocol. The measurements presented here correspond to internal friction, i.e. shearing inside the material.

Preparation included sieving (sieves specified in table 1) from 30 cm height into a shear cell of type No. 1. Measurements have been done at normal loads (normal stress) of 500, 1000, 2000, 4000, 8000, and 16.000 Pa. A shear velocity of 30 mm/min was imposed. Normal and shear load, velocity and lid displacement (volume change) were measured at 100 Hz and then down sampled to 5 Hz.

During the measurement the material is sheared for 3 minutes until a plateau is reached indicating shear has localized into a shear zone. The sample is unloaded by reversing rotation and immediately re-sheared for 3 minutes simulating reactivation of an existing shear zone.

Laboratory conditions were air conditioned during all the measurements (Temperature: 23°C, Humidity: 45%).

| GFZ-ID | CDUT-ID | Material | Sieve | File name |
|--------|---------|----------|-------|-----------|
| 373-01 | DB2017-X1_40-60_0.3-0.45mm | Qtz-sand | Fast/coarse | 373-01_CDUT-X1 |
| 374-01 | DB2017-X2_40-60_0.2-0.3mm | Qtz-sand | 250 ml/min (Geomod) | 374-01_CDUT-X2 |
| 377-01 | DB2017-L1_40-60_0.3-0.45mm | Qtz-sand | Fast/coarse | 377-01_CDUT-L1 |
| 378-01 | DB2017-L2_60-80_0.2-0.3mm | Qtz-sand | 250 ml/min (Geomod) | 378-01_CDUT-L2 |
| 379-01 | DB2017-B1_40-60_0.3-0.45mm | Glassbeads | 250 ml/min (Geomod) | 379-01_CDUT-B1 |
| 380-01 | DB2017-B2_60-80_0.2-0.3mm | Glassbeads | 250 ml/min (Geomod) | 380-01_CDUT-B2 |

*Table 1: Sample overview*

**2. Analysis method**

From the resulting shear stress curves (see e.g. Figure 1) three characteristic values (strengths) have been picked manually:

(1) The shear strength at ***peak friction*** corresponding to the first peak in the shear curve reflecting hardening-weakening during strain localization
(2) the shear strength at ***dynamic friction*** corresponding to the plateau after localization and representing friction during sliding
(3) the shear strength at ***reactivation friction*** corresponding to the second peak corresponding static friction during reactivation of the shear zone.

Matlab-based regression analysis of these friction data by means of calculating all possible two point slopes (friction coefficient) and intercepts (cohesion) for mutually combined data pairs of shear strength and normal load. These data (i.e. friction coefficients and cohesions) are then evaluated by means of univariate statistics by means of calculating mean and standard deviation and comparing the probability density function (pdf) to that of a normal distribution (see e.g. Figure 2).

Two Matlab scripts "RSTshow.m" and "RSTanalysis.m" is provided along with this data set allowing analyzing and visualizing the data.

**3. File description:**

For each material sample there exists

- (i)       shear curve data (txt format, example Table 2)
- (ii)     shear curve plot (pdf format, example Figure 1)
- (iii)    friction data (txt format, example Table 3)
- (iv)    friction plots (pdf format, example Figure 2).

An overview of all files of the data set is given in the **List fo Files.**

**3.1 Shear curve data** are given as (i) time series (ts) data in ascii format ("File name_ts.asc") and visualized as (ii) shear stress versus displacement plots ("Filename_sc.pdf"). A matlab script "RSTshow.m" is provided to reproduce the plots from the data.

| % time (s) | normal load (Pa): 500 | 1000 | 2000 | … |
|------------|----------------------|--------|-------|---|
| 0.0000 | -0.046 | -0.025 | 0.138 | |
| 0.0002 | | | | |
| … | | | | |

***Table 2: Example of shear curve time series data.*** *First line is header. First column is time in seconds (5 Hz). Columns 2-19 are shear stress (in Pa) for corresponding normal loads as specified in the header of the respective columns (6 load levels from 500 to 16.000 Pa, three repetitions each load level).*

[Figure]

***Figure 1: Example of shear curve plot.*** *Y-axis is shear stress, x-axis is displacement. Each data set consist of 18 shear curves corresponding to 6 levels of normal load with 3 repetitions each load level.*

**3.2 Friction data** are given as (iii) data pairs (shear strength and normal load) for peak, dynamic and reactivation friction in txt format ("File name_peak.txt, File name_dynamic.txt, File name_reactivation.txt"). They are visualized by (iv) plotting into Mohr Space (normal load vs. shear strength) and histograms for friction coefficients and cohesion ("File name_peak.pdf, File name_dynamic.pdf, File name_reactivation.pdf").

| %normal load [Pa] | shear strength [Pa] |
|---|---|
| 500 | 413.10 |
| 1000 | 830.39 |
| … | |

***Table 3: Example of friction data.*** *First line is header. First column is normal load. Second column is shear strength. 19 rows in total.*

[Figure]

***Figure 2: Example of friction plot.*** *Upper panel = Plot of all data pairs in the Mohr space (normal load vs. shear strength), lower panel = histograms of mutual two-point regression results for slope (friction coefficient) and y-axis intercept (cohesion). Red curve is a synthetic normal distribution with the same mean and standard deviation as the data set for comparison.*

**Cited reference:**

Schulze, D. (1994), Entwicklung und Anwendung eines neuartigen Ringschergerätes. Aufbereitungstechnik 35 (10), 524-535.

Schulze, D. (2003) Time- and velocity-dependent properties of powders effecting slip-stick oscillations, Chemical Engineering & Technology, 26, 1047-1051.

Schulze, D. (2008): Powders and Bulk Solids - Behavior, Characterization, Storage and Flow, Springer Berlin Heidelberg New York, ISBN 978-3-540-73767-4, 511 pp.

Klinkmüller, M., G. Schreurs, M. Rosenau, and H. Kemnitz (2016), Properties of granular analogue materials: A community wide survey, Tectonophysics, 666, doi.: 10.1016/j.tecto.2016.01.017.

Ritter, M., K. Leever, M. Rosenau, and O. Oncken (2016): Scaling the Sand Box - Mechanical (Dis-) Similarities of Granular Materials and Brittle Rock, J. Geophys. Res - Solid Earth, doi.: 10.1002/2016JB012915.

**begin with description/ abstract from the DOI Landing Page**

Content wish list – the further information may be different between the different data types, the list below can be regarded as "wish" list/ proposed content and is not meant to be mandatory for each data publicatoin:

**The general aim is to describe the data sufficiently to allow others working with the data without requiring additional information from the author (you).**

**Whenever necessary, we are providing a standardised data description with the content above (mandatory) that is followed by the parts below. This data description will be made available via the DOI landing page (for direct download) and also included in the data (e.g. zip folder).**

- **File inventory**: please provide a table with the file name, format and description of all files/folders → to allow people easily understand the files of the data publication. This may also be provided as in the example below
- **Tables**: please define the table contents, ideally via a table with the following columns: (1) abbreviation of the column heads; (2) unit or notation rule (e.g . for data dd-mm-yyyy or mm/dd/yyyy); (3) Description of the field
- **Figures/ Images/ Photos**: are welcome, if helpful for understanding a model setup or provide an example of the file content..
- **References**: in case there are citations in the description

---

## Author Comment (AC2) · 14 Aug 2018

Fig.10 Natural example of the Longmenshan fold-thrust-belt.    (a) Topography of Longmenshan fold-thrust-belt with the thickness of Lower Cambrian Qiongzhushi shale.  Inset map shows location.  (b) Topographic profiles across northern, central and southern segments of the Longmenshan, showing much higher topography and slope in the southern and central segments of Longmenshan.  (c-e) Cross-sections across the northern, central and southern segments of the Longmenshan, indicating much stronger deformation and shortening occurred in the basal detachment across its northern segment.

[Figure]

Please also note the supplement to this comment:
https://www.solid-earth-discuss.net/se-2018-45/se-2018-45-AC2-supplement.pdf

———————————————————

[Figure]

**Supplement:**

[Figure]

(A) Elevation 10 m — 6000 m. Songpan-Ganzi Belt, Longmen Mountains, Sichuan Basin, Chengdu, Nanchong, Ya'an, Tibetan Plateau. Thickness of shale contours: 250, 750, 1250.

Legend
- 250 — Thickness of shale (m)
- DEM sections
- Seismic sections
- Borehole

(B) DEM section-3: Min Shan, Longmen Mts., Western Sichuan Foreland Basin, SE
DEM section-2: Xuelongbao, Longmen Mts., Western Sichuan Foreland Basin, Longquan Mts., SE
DEM section-1: Longmen Mts., Western Sichuan Foreland Basin, Xiongpo Mts., Longquan Mts., SE
Elevation (km) axis. Distance (km): 0–200.

(C) Depth (km), TJ-1, NW. Units: J-K, T₃x, T₁/₂l, P₁-T₁, Z-P₂, AnZ.

(D) SE, LS1, CY9. Units: J-K, T₃x, T₂x, T₁m-T₃x, P₁-T₁, Z-P₂, AnZ.

(E) Songpan-Ganzi belt | Longmenshan fold-thrust-belt | Western Sichuan foreland basin
Depth (km), NW — SE. Klippen. Seismic line: Line LHS-02-12.
Faults: Longdong fault, Maowen-Wenchuan fault, Beichuan-Yinxiu fault, Anxian-Guanxian fault. Pre-Cambrian.
Scale: 0 10 20 km

Legend:
- Quaternary
- Cretaceous
- Jurassic
- Upper Triassic
- Low-Mid. Triassic
- Permain
- Silurian-Carboniferous
- Camrbian-Ordovician
- Pre-Cambrian

---

## Referee Comment (RC1) · Anonymous Referee #1 · 15 Aug 2018

In this manuscript, the authors discuss the influence of material properties on the development of analogue accretionary wedges. While this manuscript has potential to interest the readers of this journal, it needs some substantial improvements, mostly in terms of the form.

1. The text needs to be carefully proofread and the English needs substantial improvements. There are many problems with the grammar and syntax, as well as with some terms and wording.

2. The Description of the Apparatus (Section 3.1) should be moved to Section 2 and the Results should be presented as a separated section. It would also be good to have

a scheme with the design of the apparatus, and if possible, it would be advantageous to have more photos of the experimental results at different time steps. I understand that the slices could only be made in the final step of the experiments, but it would be nice to have some photos with the evolution of the accretionary wedges.

3. There is a considerable amount of Discussion in the description of the Results. The different experiments are described and compared together along the way in the Results section. The authors should try to make an effort to first describe the main particularities of each experiment and then make a direct comparison between them.

---

## Short Comment (SC2) · 15 Aug 2018

Correct citation and doi for the data publication:

Deng, Bin; Rosenau, Matthias; Schönebeck, Jan (2018): Ring-shear test data of rock analogue materials from Chengdu University of Technology (EPOS Transnational Access Call 2017). V. 1. GFZ Data Services. http://doi.org/10.5880/GFZ.4.1.2018.003

Please include the citation in the reference list and cite it in the text as "Deng et al. (2018)".

Please report the citation also in "Assetts".

[Figure]

Please include the sentence "Ring shear test data are published open access in Deng et al. (2018)" > in line 86 (section 2.3) as well as > in captions of figure 2 and 3.

Please include the sentence "We thank GFZ Data Service for publishing the ring shear test data" in the Acknowledgements.

Please change text in lines 378-380 to "Ring shear tests were performed as a remote service of GFZ Potsdam in the frame of EPOS (European Plate Observing System) Transnational Access activities."

Until registration of the doi, please use the review link for preview:

http://pmd.gfz-potsdam.de/panmetaworks/review/7b2357724c023eeb9c307d51be83d70876528604ccac41b90ed436ec14

---

## Author Comment (AC3) · 25 Aug 2018

Following the advice from anonymous Referee #1, we have added figures to show apparatus, and structural evolution of setup 1-6. Fig.1 (a) Diagram of the experimental apparatus. (b) Interpreted fold-and-thrust wedge showing the parameters measured in the experiment results. Fig.2 Sequential photographs showing the structural evolution of Setup-1. Fig.3 Sequential photographs showing the structural evolution of Setup-2. Fig.4 Sequential photographs showing the structural evolution of Setup-3. Fig.5 Sequential photographs showing the structural evolution of Setup-4. Fig.6 Sequential photographs showing the structural evolution of Setup-5. Fig.7 Sequential photographs

showing the structural evolution of Setup-6.

Please also note the supplement to this comment:
https://www.solid-earth-discuss.net/se-2018-45/se-2018-45-AC3-supplement.pdf

———————————————————

[Figure]

**Supplement:**

[Figure]

a)

End wall
Moving wall

800 mm
700 mm

Glass side walls

10 mm
10 mm
15mm

Motor

White quartz sand    Color quartz sand    PIV area

b)

Frontal-deformation zone    Frontal-imbrication zone    Internal-accumulation zone

$\alpha$

$T_5$    $T_4$    $T_2$    $T_1$

$T_6$    $T_3$

$H$

$H_0$    $D_n$

$\theta (T_5)$    $L$

Measured marker layer

$H_0$= Initial height    H= Wedge height    Tn= Major fault    Dn= Fault displacement    L= Length of wedge

$\alpha$= Wedge slope angle    $\theta$ (Tn)= Thrust ramp angle

a) 15 mm

b) 65 mm

c) 90 mm

d) 135 mm

e) 190 mm

f) 235 mm

g) 275 mm

h) 320 mm

i) 400 mm

j) Interpretation

strength (Pa)
50   150
depth (cm)

0    5    10 cm

Setup 1:Quartz sand (0.3~0.45mm)

[Figure]

a) 10 mm

T₁

b) 45 mm

T₂ T₁

c) 75 mm

T₃    T₂ T₁

d) 145 mm

T₄    T₃    T₂ T₁

T₃₋₁

e) 180 mm

T₄    T₃    T₂ T₁

f) 250 mm

T₅    T₄    T₃    T₂ T₁

g) 300 mm

T₅    T₄    T₃    T₂ T₁

h) 380 mm

T₆    T₅    T₄    T₃    T₂ T₁

i) 400 mm

T₆    T₅    T₄    T₃    T₂ T₁

strength (Pa)
50   150
depth (cm)
0
1
2

j) Interpretation

T₆    T₅₋₁ T₅₋₂ T₄    T₃    T₃₋₁ T₂ T₁

0    5    10 cm

Setup 2: Quartz sand + Glass beads (1 mm)

[Figure]

a) 10 mm

b) 40 mm

c) 75 mm

d) 115 mm

e) 185 mm

f) 245 mm

g) 325 mm

h) 355 mm

i) 400 mm

j) Interpretation

strength (Pa)

depth (cm)

0   5   10 cm

Setup 3: Quartz sand + Glass beads (3 mm)

[Figure]

a) 20 mm

b) 60 mm

c) 75 mm

d) 105 mm

e) 165 mm

f) 190 mm

g) 235 mm

h) 300 mm

I) 370 mm

j) 400 mm

k) Interpretation

strength (Pa)

depth (cm)

0    5    10 cm

Setup 4: Quartz sand (0.2~0.3 mm)

[Figure]

a) 25 mm

b) 85 mm

c) 145 mm

d) 170 mm

e) 230 mm

f) 250 mm

g) 320 mm

h) 350 mm

i) 400 mm

j) Interpretation

strength (Pa)
50    150
0
depth (cm)
1
2

0    5    10 cm

Setup 5: Quartz sand + Glass beads (1 mm)

[Figure]

a) 5 mm

b) 30 mm

c) 60 mm

d) 85 mm

e) 150 mm

f) 170 mm

g) 270 mm

h) 295 mm

i) 400 mm

j) Interpretation

strength (Pa)

depth (cm)

Setup 6: Quartz sand + Glass beads (3 mm)

---

## Referee Comment (RC2) · F. Rossetti (Referee) · 6 Sep 2018

Dear Authors,

I have received two independent reviews (one uploaded in the system, the other attached below) of your submitted ms. Despite the research topic is relevant since it is thought to improve the state of knowledge regarding the physical properties that control the evolution and mechanical behaviour of accretionary wedges at convergent plate margins, both reviewers indicate that much work is needed to render the manuscript suitable for publication. I concur with this evaluation.

[Figure]

In addition to comments made by the reviewers and by Dr. Roseneau (see SC1 and SC2), I would note these major points that should merit consideration when preparing a revised version of the manuscript: (i) The scope of the manuscript and the geological problem is not fully described/introduced. A vast literature exists regarding the structural, mechanic and dynamic controls on the accretionary wedge evolution at convergent plate margins. The idea to develop new achievements regarding the state-of-the-art dealing with the mechanics of accretionary wedges is thus relevant and challenging at the same time. Therefore, the scope of the study and its relevance should be made more explicit; (ii) Existing literature on the analogue and numerical modelling of thrust and fold belts and accretionary wedges is not adequately quoted. Fundamental papers on the mechanics of the thrust and fold belts (introducing the "critical taper" theory), such as Davis et al. (1983, JGR), Dahlen et al. (1984, JGR), Dahlen (1990, Ann. Rev. Earth Planet Sci) are not quoted; (iii) As it stands, the manuscript appears as poorly organised, often mixing data presentation with inferences and presenting data in the discussion section. Consequently, the presentation of the experimental results appears to be rather chaotic and non-systematic; (iv) Results should not be compared only to nature, rather (and above all) to existing analogue and numerical modelling studies [see e.g. Morgan (2015, JGR), Gray et al. (2014, J Struct Geol), Ruh et al (2012, Tectonics), Simpson (2011, Tectonophysics), Bose et al. (2009, J Struct Geol), Gutscher et al (1998 , JGR)] and critically discussed. I would emphasise that the comparison with the natural cases appears, at least for the Zagros case, too simplistic, since ductile rheologies should be also taken into account. As general remark, your experiments suggests that an in-depth strength drop affects model evolution and style of deformation. This should be compared, for example, with the existing literature models dealing with brittle-viscous rheologies rather than simply to natural cases. Furthemore, a more deep analysis of the implications of your results in terms of the critical taper theory could increase the significance and the potential impact of the presented results. Potentially relevant (but not fully made explicit) are also the outputs of this study in terms of the structural styles of the accretionary wedges in nature. In other terms,

broader impact of the presented research results is largely (and critically) depending on how the outcomes of the sensitivity analysis of main physical parameters investigated in this study can be extrapolated to improve the knowledge of the main physical properties that control orogenic wedge evolution at convergent plate margins.

Finally, despite I am not a native English speaker, I found the text rather rough in some parts. Therefore a revision of the English text is necessary to improve readability of the manuscript.

I have uploaded an annotated version of the submitted pdf file where the above points are further detailed.

Based on the above, since the required amount of revision is a major one and a general re-focusing and re-organisation of the manuscript is necessary, my decision is to reject the manuscript at this stage.

Sincerely, Federico Rossetti

REVIEWER#2 (formal review)

Dear editor,

In their manuscript entitled "What degree the geometry and kinematics of accretionary wedges in analogue experiments is dependent on material properties", Jiang et al., report on a study consisting of a set of analogue experiments on accretionary wedge formation, for which more or less the same set-ups but different materials have been used. Further, the thickness of the basal detachment was varied. Experiments were conducted in the "push mode".

The experiments of this study are very similar to each other and parameters have been tested the variation of which will not result in large but potentially significant differences. Therefore, physical properties of the analogue materials used (sand with two different grain size spectra, micro glass beads) need to be known with high accuracy, experimental set-ups need to be extensively described, and observations need to be thoroughly discussed.

Several parameters like grain size and aspect ratio, bulk density, friction coefficients and cohesion have been measured e.g. from SEM photos or with the ring-shear tester at GFZ, Potsdam. Whereas Table 1 provides a good overview about the results of the measurements, the table lacks the presentation of errors. For the frictional parameters, these are included in the figure showing measurement results and regression lines, but lacking at all otherwise. Although the authors stress that besides the impact different materials with different properties have on the evolution of experimental wedges they are interested in the "human factor", they do not provide any information on how often they repeated their measurements. Similarly, they do not report on temperature, humidity, and other properties they mention in their introduction. Cohesion has been found to be a parameter difficult to estimate. However, the authors do not report on repeated measurements and data processing. It is very confusing that they present a certain range of each parameter when describing their materials while it is not clear if they talk about the variability among various materials.

The experimental set-up is described, but a figure is lacking. What kind of lubrication was used? There were two experiments with only sand – but what was the material the bottom of the experimental box was made of? There will be always a "detachment" as with pushing the backwall you force the material to detach. Measuring angles from cross-sections is not very accurate – what was the error here? Can you really measure angles as accurate as $0.1°$? Which points did you use to measure the taper? This is important as the authors intend to identify the impact of rather small differences of material properties on wedge growth. Have the experiments been repeated?

When describing the results of the experiments in terms of how the wedges looked like after 40 cm of shortening, the authors already interpret differences they find – this is misplaced as it should be a theme of the discussion. When describing your results, please stay neutral! You say that your wedges grew self-similarly – however, you do not show any sequence of images. And keep in mind, that a wedge needs to consists

of at least a few thrust slices before if behaves according to the critical taper concept and that some parts of a wedge are not in a critical state (e.g. Lohrmann et al., 2003). Thus, your arguments would be much stronger if you had applied more shortening. You very often say that the wedge reached a critical slope – how do you know without having performed a critical taper analysis?

A main difference between your experiments "without a detachment" as you call it, and the experiments which have a thin basal layer made of glass beads is the in the latter experiment the basal detachment has some topography and will evolved in the weaker material. Thus, you need to be very cautious when comparing your experiments.

The role of the basal detachment has been addressed many times before, e.g. Gutscher et al., 1996; Contardo et al., 2011, . . . Thus, thorough discussion is possible and relevant studies need to be cited.

Analysis of the experiments presented here seems to be restricted on purely geometrical parameters. Therefore, you cannot claim anything about kinematic evolution – to do so you need to monitor fault activity. Is there any difference between the temporal evolution of fault activity and fault reactivation between the experiments of this study?

A sufficient size of the sample is crucial when performing statistical analyses – is this the case here? I doubt this.

Basically, studies focusing on very thoroughly analyzing experimental wedges and elucidating the impact of relatively small parameter variations and relatively little varying set-ups etc. are very important as they provide insight into the relative importance of potential control parameters, the robustness of experimental set-ups, etc. Such studies offer data to better understand the similarities, differences, and variability between natural wedges. Further, such studies are valuable as they also confirm the feasibility of an experimental set-up. Thus, the present study would be of importance, however, the present version of the manuscript is not at all ready for publication. Several additional analyses should be performed as indicated above. Further, experiments need to be described in more detail. Errors should be quoted with much more cautiousness. The same, cautiousness is a pre-requisite when stating e.g. that a wedge is in a critical state. To do so without a proper CT-analysis is a no go.

How to proceed? I strongly recommend to reject this manuscript in its present version and ask the authors to use the comments and reviews they get to improve their study. This will in any case mean they should perform more analyses, and may also mean they need to repeat some experiments and apply more shortening. However, this effort should be worthwhile as then they could come up with a proper manuscript. As it was quite difficult to read the manuscript due to the poor language, I also highly recommend proofreading by a native speaker.

Please also note the supplement to this comment:
https://www.solid-earth-discuss.net/se-2018-45/se-2018-45-RC2-supplement.pdf

**Supplement:**

[revised manuscript text omitted]

Topgraphy along cross section after: 0 mm, 100 mm, 200 mm, 300 mm, 400 mm, bulk shortening.